# Visual Attention-Prompted Prediction and Learning

## Abstract

Explanation(attention)-guided learning is a method that enhances a model's predictive power by incorporating human understanding during the training phase. While attention-guided learning has shown promising results, it often involves time-consuming and computationally expensive model retraining. To address this issue, we introduce the attention-prompted prediction technique, which enables direct prediction guided by the attention prompt without the need for model retraining. However, this approach presents several challenges, including: 1) How to incorporate the visual attention prompt into the model's decision-making process and leverage it for future predictions even in the absence of a prompt? and 2) How to handle the incomplete information from the visual attention prompt? To tackle these challenges, we propose a novel framework called Visual Attention-Prompted Prediction and Learning, which seamlessly integrates visual attention prompts into the model's decision-making process and adapts to images both with and without attention prompts for prediction. To address the incomplete information of the visual attention prompt, we introduce a perturbation-based attention map modification method. Additionally, we propose an optimization-based mask aggregation method with a new weight learning function for adaptive perturbed annotation aggregation in the attention map modification process. Our overall framework is designed to learn in an attention-prompt guided multi-task manner to enhance future predictions even for samples without attention prompts and trained in an alternating manner for better convergence. Extensive experiments conducted on two datasets demonstrate the effectiveness of our proposed framework in enhancing predictions for samples, both with and without provided prompts.

## 1 Introduction

Deep learning algorithms have achieved exceptional performance in computer vision and have been applied widely in many areas (Erickson et al., 2017); However, Deep Neural Networks (DNNs) cannot ensure the reasonableness of AI's decision due to its "black box" structure (Adadi & Berrada, 2018). Explainable AI (XAI) has gained significant importance in recent years as a means to comprehend the underlying rationale behind a model's decision-making process. Currently, most existing work, such as CAM (Zhou et al., 2015), Grad-CAM (Selvaraju et al., 2017), and integrated gradients (Qi et al., 2019), primarily generates saliency or attention maps as explanation maps. These maps highlight areas in input data (e.g., an image) that the model emphasizes when making a decision.

Despite considerable progress in XAI, explaining the model reasoning process is not our ultimate goal. What is more crucial is understanding how XAI can guide DNNs to enhance both the correctness of the model's reasoning process and prediction output. However, certain challenges remain underexplored to achieve it: How to identify the wrong model reasoning process and more importantly, how to correct the wrong model reasoning? To address those problems, explanation(attention)-guided learning is introduced. Chen et al. (2020); Gao et al. (2022c); Shen et al. (2021); Gu et al. (2023) use human-annotated attention map to steer the reasoning process of DNNs for image classification. Hsieh et al. (2023); Raffel et al. (2020); Narang et al. (2020) extract rationales as additional supervision besides label supervision for training large language models (LLM). Attention-guided learning aims to train a model such that its reasoning is corrected based on the training data. These works primarily focus on simultaneously minimizing the prediction error and reducing the discrepancy between the model's reasoning and the true reasoning in the training set. This is primarily achieved by reducing

the difference between the model-generated attention area and the ground truth attention map, which serves as a surrogate for true reasoning. In many real-world situations, when new samples with prompts are available, we desire the model to directly make predictions using a prompt that informs it of the rationale behind its decision, as illustrated in Figure 1 (a). However, attention-guided learning falls short of meeting this need. As depicted in Figure 1 (b), it necessitates retraining the model with all samples, which is both time-intensive and computationally costly.

Although many works about making predictions with prompts have been conducted such as (Oymak et al., 2023; Liu et al., 2023; Ye et al., 2022), they mainly focus on text data with no work designed for the applications to image data to our best knowledge. Existing attention-prompted prediction can not be directly adapted to handle image data because of the following challenges: **1) How to incorporate the visual attention prompt into the model's prediction process and leverage it for future predictions even without prompts?** Traditional Convolutional Neural Network (CNN)–based classifiers rely exclusively on images for decision-making, lacking a prompting mechanism that integrates user-provided attention guidance. Furthermore, prompts may not always be available for many

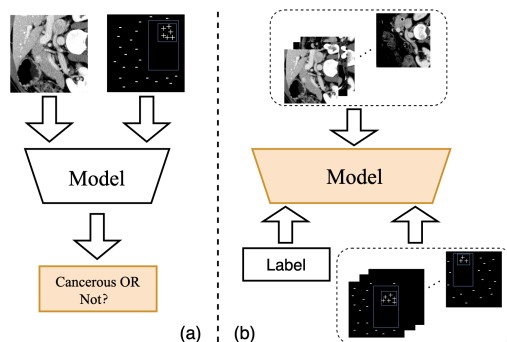

Figure 1: Illustration of (a) direct prediction using a prompt and (b) attention-guided learning. The symbol '+' indicates an important region, while '-' denotes an unimportant region.

samples, complicating the model's ability to predict using both images with and without prompt guidance. **2) How to handle incomplete information from visual attention prompts?** Visual attention prompts often contain incomplete information because of the user's variance in the level of domain expertise and cost of precise prompting (Karimi et al., 2020). In many cases, it's straightforward for users to identify which regions are important and which are not, yet other areas may remain ambiguous for an image. Introducing such prompts with incomplete information into the model for decision-making presents a significant challenge.

To address the above challenges, we propose a Visual Attention-Prompted Prediction and Learning framework that integrates visual attention-prompt guidance into the decision-making phase. This work makes several contributions, which can be summarized as follows:

- A framework designed to incorporate visual attention prompts in the model decision-making process. The framework can adapt to images both with and without attention prompts for prediction.

- A regularized two-stream CNN architecture designed to preserve prompt information and enhance future predictions. This structure facilitates knowledge sharing between CNNs, allowing future samples without prompts to benefit from images with prompts during training.

- A newly proposed attention prompt modification method that handles prompts with incomplete information. This method utilizes a perturbation-based model to gauge the importance of each pixel. It then optimizes the initially provided attention prompt map by aggregating randomly perturbed masks weighted by the importance scores.

- An optimization-based mask aggregation method. We propose a monotonic and endpoint-preserved multilayer perceptron (MLP) to learn the importance scores of each perturbed mask, leveraging the probability scores computed by the classifier, thereby achieving adaptive attention aggregation.

- Comprehensive experiments are conducted using a variety of prediction evaluation metrics across two datasets. The results underscore the effectiveness of our proposed framework in improving model predictability for images, both with and without visual attention prompts.

## 2 RELATED WORK

**Attention-guided Learning** The integration of human knowledge into interpretable models has been extensively studied in NLP and tabular data through techniques such as attribution and feature regularization (Gao et al., 2022a;b). Recently, there has been an increasing awareness of the importance of visual attention maps. One popular approach is to obtain local attention maps that highlight the input features most responsible for a model's prediction, such as Grad-CAM (Montavon et al., 2019;

Selvaraju et al., 2017). These attention maps have been used as supervision signals aligned with prediction loss, in order to further improve model performance (Shen et al., 2021; Gao et al., 2022c). HAICS (Shen et al., 2021) is a conceptual framework for image classification with human annotation in the form of scribble annotations as the attention signal. To alleviate the problems inherent within human labels, Gao et al. (2022c) develops a novel objective that handles inaccurate, incomplete, and inconsistent distribution of human annotation. However, most existing works typically require a large volume of attention maps, which are costly to obtain.

**Attention Prompt** The prompting technique originates from NLP (Devlin et al., 2018) and is transferred to CV (Dosovitskiy et al., 2020). Oymak et al. (2023) explore prompt-tuning for one-layer attention architectures and study contextual mixture models where each input token belongs to a context-relevant or -irrelevant set on large language models (LLM). VPT (Jia et al., 2022) adds multiple adaptable prompt tokens to the patch tokens of the static ViT model, it adapts to a range of downstream tasks and datasets without requiring a complete fine-tuning of the ViT model. Paiss et al. (2022) propose a novel explainability-based approach that enhances one-shot classification recognition rates by adding a loss term to ensure CLIP addresses all pertinent semantic input aspects. Li et al. (2023) introduce saliency prompts which are generated by saliency masks indicating potential objects, and then are used to decorate the kernels for injecting location and shape knowledge.

**Multi-task Learning** Multi-task learning refers to a training process where machine learning models are trained with multiple tasks simultaneously, using shared representations to learn across related tasks (Crawshaw, 2020). Existing MTL methods can be generally categorized into two groups: hard parameter sharing and soft parameter sharing. Hard-parameter sharing in deep neural networks often requires sharing model weights between multiple tasks to jointly minimize multiple loss functions (Ouyang et al., 2014; Zhang et al., 2014). Typically, existing methods share lower-level layers for representation learning, while keeping a few task-specific layers to specialize in distinct tasks (Crawshaw, 2020). Alternatively, soft-parameter sharing methods offer greater flexibility, where each task has its own dedicated models. Regularization techniques are employed to enforce task relatedness by adding distance between the parameters of task models as training objective (Duong et al., 2015; Misra et al., 2016; Yang & Hospedales, 2016). Some studies have also extended soft-parameter sharing by incorporating Neural Architecture Search (Elsken et al., 2019), learning neural network modules jointly with task ordering (Liang et al., 2018; Meyerson & Miikkulainen, 2017). Aiming for the conciseness of hard-parameter sharing and the flexibility of soft-parameter sharing, recent research has emphasized sharing representation learning layers while leveraging task relationships in task-specific layers (Long et al., 2017; Strezoski et al., 2019; Bai & Zhao, 2022).

## 3 METHODOLOGY

In this section, we begin by presenting the problem formulation. We then introduce our overall framework for visual attention-prompted prediction, followed by the proposed attention prompt map modification method with an adaptive mask aggregation model. Lastly, we present our model learning strategy that incorporates attention-prompt guided multi-task learning and alternating training.

### 3.1 PROBLEM FORMULATION

Given a sample pair $(I, D, y)$, where $I \in \mathbb{R}^{C \times H \times W}$ represents the original image, with $C$, $H$, and $W$ denotes the number of channels, height, and width, respectively. The class label of the original image is represented by $y \in \mathbb{R}$. The visual attention prompt for image $I$, corresponding to its class label $y$, is denoted by $D \in \mathbb{R}^{H \times W}$, with dimensions identical to the original image but with one channel. Attention prompt $D$ serves as an instruction or signal indicating which parts of the input image are particularly relevant or should be prioritized. The goal of prediction with the attention prompt is to use it to guide the process of mapping an image to its class label. Formally, given an image and its attention prompt, this process can be represented as $(I, D) \mapsto y$, where $D$ is optional.

### 3.2 PROPOSED FRAMEWORK

To address the challenges stated above and incorporate attention prompts into the model decision-making process, we introduce the Visual Attention-Prompted Prediction and Learning framework, depicted in Figure 2. This framework can take images with and without attention prompts as input.

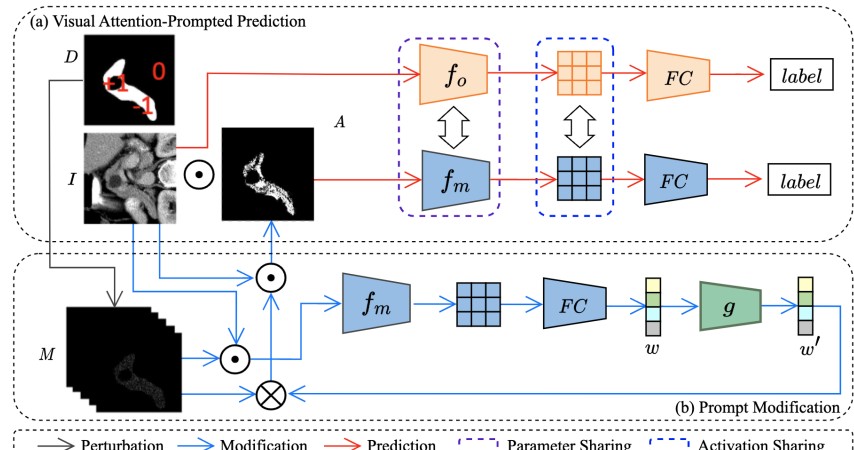

Figure 2: Illustration of the proposed Visual Attention-Prompted Prediction and Learning Framework: Subfigure (a) depicts our proposed two-stream CNN architecture, learned in a prompt-guided multi-task manner, while Subfigure (b) outlines the attention optimization process through prompt-guided perturbation.

Specifically, as illustrated in subfigure (a), we instantiate a two-stream CNN architecture with two identical CNNs as classifiers: 1) An original-input classifier, denoted as $\mathbf{f_o}$, which directly predicts the class label from an image in the absence of a visual attention prompt, and 2) A prompted-input classifier, denoted as $\mathbf{f_m}$, which takes the image with a visual attention prompt provided. When processing input images with an attention prompt, the input image undergoes masking based on the attention prompt, retaining only a subset of pixels corresponding to the relevant regions. In cases where attention prompts may lack complete information, we introduce an attention prompt map modification method denoted as $\mathbf{t}(\cdot)$, as illustrated in subfigure (b). This method is based on aggregating randomly perturbed masks using an optimization-based weight learning function that is learned on the prediction confidence scores from classifiers. Detailed information will be presented in Section 3.4. Formally, the prediction process of our proposed framework can be expressed as:

$$y \leftarrow \begin{cases} \mathbf{f_m}(I \odot \mathbf{t}(D)), & \text{if both } I \text{ and } D \text{ is provided} \\ \mathbf{f_o}(I), & \text{if only } I \text{ is provided} \end{cases} \tag{1}$$

where $\odot$ denotes element-wise multiplication, the details of learning will be mentioned in Section 3.5.

### 3.3 Perturbation-based Attention prompt map modification

As stated above, the information from the attention prompt is frequently incomplete in real-world scenarios. To tackle this issue, we introduce a perturbation-based method that modifies the visual attention prompt and enables more effective prompt guidance during the image classification process.

Suppose we have a visual attention prompt $D \in \mathbb{R}^{H \times W}$, which is an $\{-1, 0, +1\}$ valued mask for each pixel $\lambda$ where $\lambda = 0$ indicates that the pixel is unimportant, $\lambda = +1$ signifies the pixel is important, and $\lambda = -1$ denotes that the importance of the pixel is unclear. First, we randomly perturb $D$ to generate $N$ binary masks $\boldsymbol{M} = \{M_i\}_{i=1}^{N}$ by setting each pixel with a value equal to $-1$ to $+1$ with probability $p$ and to $0$ otherwise. More specifically, we define $\lambda$ as a pixel of $M_i$ as well as the pixel of $D$ with the corresponding coordinate, and $p'$ as a number uniformly randomly sampled from $[0, 1]$. Consequently, we can obtain the perturbed mask $M_i$ as follows:

$$M_i(\lambda) = \begin{cases} +1, & \text{if } p' \geq p \text{ and } D(\lambda) = -1 \\ 0, & \text{if } p' < p \text{ and } D(\lambda) = -1 \end{cases} \tag{2}$$

After obtaining $N$ perturbed masks, we proceed to modify the initial attention prompt masks that elucidate the decision of model $\mathbf{f_m}$ on images. First, we perform an element-wise multiplication on the original image and each randomly perturbed mask to obtain the masked image. Subsequently, we compute the confidence scores $\boldsymbol{w} = \{w_i\}_{i=1}^{N}$ of each mask in $\boldsymbol{M}$. Formally, the confidence score computation process can be summarized as:

1. Randomly perturb the initial visual attention prompt map $D$ according to Equation 2 to obtain perturbed masks $\boldsymbol{M} = \{M_i\}_{i=1}^{N}$.

2. Perform element-wise multiplication of the image and each perturbation mask to obtain images masked by each mask $\{(I \odot M_i)\}_{i=1}^{N}$.
3. Compute the confidence score of masked images corresponds to its class label $w_i = \mathrm{softmax}_k \mathbf{f_m} (I \odot M_i), i = 1, \ldots, N$, where $k$ represents the index of the label $y$ and $\mathrm{softmax}_k$ denotes the output of the softmax layer corresponding to the $k$-th class.

Upon obtaining the confidence score for each perturbed mask, we proceeded to aggregate these masks by weighted averaging using the computed confidence score as weights. We then normalize it by the expectation of each pixel $N \cdot p$. This process yields the final modified attention prompt mask $A$, effectively synthesizing the pertinent information derived from the various masks and their corresponding weights. To summarize, the above computation process can be represented as:

$$A = \frac{1}{N \cdot p} \sum_{j=1}^{N} \mathbf{f_m}(I \odot M_j) \cdot M_j \qquad (3)$$

Consequently, the modified attention prompt map is computed by aggregating randomly perturbed masks weighted by their confidence scores.

### 3.4 ADAPTIVE ATTENTION AGGREGATION METHOD

While aggregating randomly perturbed masks weighted by confidence scores is a viable approach, using the non-parametric linear combination method for aggregating is overly simplistic. This method presumes a linear relationship between the confidence scores of masks and their importance in aggregating the final explanation mask. To overcome this limitation, we propose a new optimization-based annotation aggregation technique. Given that mask aggregation depends on the confidence scores generated by the classifier, our optimization-based aggregation method integrates a weight-learning function, $\mathbf{g}(\cdot)$, to optimize these scores and ascertain the best weights for mask combination, represented as $\mathbf{g} : w \mapsto w'$. Considering the properties of the masks and their corresponding confidence scores, our weight-learning function should follow these properties:

- *(Monotonicity)*: The function should exhibit monotonic behavior, meaning it should be monotonically non-decreasing. This property ensures that masks with higher confidence scores are assigned greater importance in the mask aggregation process.
- *(Endpoint-preserving)*: The function, which takes input values ranging from 0 to 1, should uphold the endpoint-preserving property. Specifically, an input of 0 should yield an output of 0, and an input of 1 should produce an output of 1. This property guarantees that the function maintains the boundaries of the input range, preserving the significant relationship between confidence scores and weights, while minimizing the negative effects of extremely large or negative values.

To achieve the monotonicity and endpoint-preserving properties, we propose a constrained MLP structure as our weight learning function to optimize the confidence score computed by the trained classifiers. To be more specific, ensuring the monotonically non-decreasing behavior of the MLP requires us to focus on a straightforward criterion: the weight matrix must consist of non-negative entries. This crucial constraint is enforced by applying weights derived from a function with a range of positive numbers (Nguyen et al., 2023). In order to satisfy

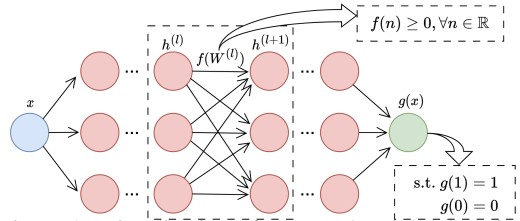

Figure 3: Visualization of proposed weights learning function based on constrained MLP.

the endpoint-preserving property, two specific measures have been taken: 1) To guarantee that the function produces an output of 0 when the input is 0, the bias term has been eliminated from all layers of the MLP. Additionally, an activation function $\sigma(\cdot)$ has been selected such that $\sigma(0) = 0$, and 2) To ensure that the function yields an output of 1 when the input is 1, a constraint term is incorporated into the training process. To be more specific, the mapping between layer $k$ and layer $k + 1$ in our proposed constrained MLP, which can be visualized as depicted in Figure 3, is as follows:

$$h^{(k+1)} = \sigma \left( \phi \left( W^{(k)} \right) h^{(k)} \right), k = 1, \ldots, L \qquad (4)$$

where $\phi$ is a function with a range in the positive numbers such as the exponential function, (element-wise) square function, or translated hyperbolic tangent, and $L + 1$ is the total number of layers.

With this newly proposed weights learning function $\mathbf{g}(\cdot)$, the perturbed masks aggregation process from Equation 3 can be reformulated as

$$A_i = \frac{1}{N \cdot p} \sum_{j=1}^{N} \mathbf{g}(\mathbf{f_m}(I \odot M_j)) \cdot M_j \tag{5}$$

Consequently, the final modified attention prompt map is derived by aggregating randomly perturbed masks using the optimized weights, as shown in Figure 2 (b).

### 3.5 Learning objectives

Our proposed framework is designed to make predictions without necessitating a user-provided visual attention prompt since such prompts might not always be available in real-world situations. To benefit samples without attention prompts, we structured our framework around attention prompt-guided multi-task learning. Additionally, we employ an alternating training strategy during model training to ensure better convergence.

#### 3.5.1 Attention prompt-guided Multi-task Learning

To ensure consistent decisions by two classifiers for the same input images, regardless of the presence or absence of an attention prompt, it is crucial that their knowledge is sharable during training. To achieve this, we adopt an attention prompt-guided multi-task learning strategy.

Specifically, we enforce a constraint ensuring similarity within the parameters of their convolutional layers. For subsequent layers, instead of imposing parameter similarity, we advocate a constraint on their feature maps before logical layers, as illustrated in Figure 2 (a). This regularization not only nudges the classifiers towards common feature extraction from both pristine and masked images—thus promoting knowledge transfer between tasks—but crucially, it diminishes the activity of hidden neurons in the original-input CNN $\mathbf{f_o}$ corresponding to dormant ones in the prompted-input CNN $\mathbf{f_m}$. Such an approach accentuates the significance of salient information spotlighted by the visual attention prompt, thereby enhancing the classification prowess of both classifiers.

Formally, the regularization is based on the distance between parameters and the last feature maps of the two classifiers $\mathbf{f_m}$ and $\mathbf{f_o}$ (Duong et al., 2015). Let $W_{\mathbf{f_m}}$ and $W_{\mathbf{f_o}}$ represent the convolutional layer parameters of two classifiers, and $S$ denote a set of model architectures before the logical layers. The two regularization term $\mathcal{L}_{\text{Param}}$ and $\mathcal{L}_{\text{Feature}}$ is formulated as follows:

$$\mathcal{L}_{\text{Param}}(\theta_{\mathbf{f_m}}, \theta_{\mathbf{f_o}}) = \|W_{\mathbf{f_o}} - W_{\mathbf{f_m}}\|_F^2 \tag{6}$$

$$\mathcal{L}_{\text{Feature}}(\theta_{\mathbf{f_m}}, \theta_{\mathbf{f_o}}) = \|S(\mathbf{f_o}(I)) - S(\mathbf{f_m}(I \odot A))\|_F^2 \tag{7}$$

Here, the first term denotes the parameter sharing regularization while the second term denotes the feature sharing regularization and $\| \cdot \|_F^2$ represents the squared Frobenius norm (Ma et al., 1994).

With the two proposed classifiers, the predicted probability for class $a$ of the $i^{th}$ data point with and without attention prompt can be written as:

$$p_m^{ia} = \text{softmax}_a \left( \mathbf{f_m} \left( I_i \odot A_i \right) \right)$$
$$p_o^{ia} = \text{softmax}_a \left( \mathbf{f_o} \left( I_i \right) \right) \tag{8}$$

Therefore, the cross-entropy between the prediction of two classifiers and the target can be written as

$$\mathcal{L}_{\text{Pred}}(\theta_{\mathbf{f_m}}, \theta_{\mathbf{f_o}}, \theta_{\mathbf{g}}) = -\sum_{i=1}^{K} \sum_{a=1}^{C} \hat{y}_{ia} \cdot \left( \log \left( p_m^{ia} \right) + \log \left( p_o^{ia} \right) \right) \tag{9}$$

where $\hat{y}_{ia}$ is the ground truth label for class $a$ of the $i^{th}$ data point in one-hot encoding. To sum up, the learning objective can be presented as follows:

$$\begin{aligned} \text{minimize} \quad & \mathcal{L}_{\text{Pred}}(\theta_{\mathbf{f_m}}, \theta_{\mathbf{f_o}}, \theta_{\mathbf{g}}) + \lambda_1 \mathcal{L}_{\text{Param}}(\theta_{\mathbf{f_m}}, \theta_{\mathbf{f_o}}) + \lambda_2 \mathcal{L}_{\text{Feature}}(\theta_{\mathbf{f_m}}, \theta_{\mathbf{f_o}}) \\ \text{subject to} \quad & \mathbf{g}(1) = 1 \end{aligned} \tag{10}$$

where $\lambda_1$ and $\lambda_2$ are the weighting hyper-parameters for parameter sharing regularization loss and feature map sharing regularization loss, respectively. If we treat the constraint as a Lagrange multiplier (Gordon & Tibshirani, 2012) and solve an equivalent problem by substituting the constraint to an MLP regularization term $\mathcal{L}_{\text{MLP}}(\theta_{\mathbf{g}})$, thus, our overall objective function can be rewritten as:

$$\min_{\theta_{\mathbf{f_m}}, \theta_{\mathbf{f_o}}, \theta_{\mathbf{g}}} \mathcal{L}_{\text{Pred}}(\theta_{\mathbf{f_m}}, \theta_{\mathbf{f_o}}, \theta_{\mathbf{g}}) + \lambda_1 \mathcal{L}_{\text{Param}}(\theta_{\mathbf{f_m}}, \theta_{\mathbf{f_o}}) + \lambda_2 \mathcal{L}_{\text{Feature}}(\theta_{\mathbf{f_m}}, \theta_{\mathbf{f_o}}) + \lambda_3 \mathcal{L}_{\text{MLP}}(\theta_{\mathbf{g}}) \tag{11}$$

where $\mathcal{L}_{\text{MLP}}(\theta_{\mathbf{g}}) = \|\mathbf{g}(1) - 1\|$, $\|\cdot\|$ is commonly chosen to be the squared $\ell_2$-norm, $\lambda_3$ is the weighting hyper-parameters for MLP regularization term.

### 3.5.2 ALTERNATING TRAINING ALGORITHM

From Figure 2, we can observe that there are three models that we need to train: the original-input CNN $\mathbf{f_o}$, the prompted-input CNN $\mathbf{f_m}$, and the constrained MLP $\mathbf{g}$. Additionally, the input of $\mathbf{g}$ depends on the output of $\mathbf{f_m}$, while the input of $\mathbf{f_m}$ relies on the output of $\mathbf{g}$. This interdependence can potentially lead to training instability as the cyclic dependencies can create a challenging feedback loop that any small changes in one model's parameters can have cascading effects on the others, making it challenging to find a stable configuration. To solve this problem, we design an alternating training algorithm to break the cyclic dependency and lead to better convergence by training models alternately.

---

**Algorithm 1:** Alternating Training

**Require:** $I, D, y$
**Ensure:** $\mathbf{f_m}, \mathbf{f_o}, \mathbf{g}$
1: **for** $t = 1 : T$ **do**
2:     **for** $q = 1 : F$ **do**
3:         Compute $\nabla_{\theta_{\mathbf{f_m}}}$ based on Equation 12
4:         Compute $\nabla_{\theta_{\mathbf{f_o}}}$ based on Equation 13
5:         $\theta_{\mathbf{f_m}} \leftarrow \theta_{\mathbf{f_m}} - \eta \nabla_{\theta_{\mathbf{f_m}}}$
6:         $\theta_{\mathbf{f_o}} \leftarrow \theta_{\mathbf{f_o}} - \eta \nabla_{\theta_{\mathbf{f_o}}}$
7:     **end for**
8:     **for** $q = 1 : G$ **do**
9:         Compute $\nabla_{\theta_{\mathbf{g}}}$ based on Equation 14
10:       $\theta_{\mathbf{g}} \leftarrow \theta_{\mathbf{g}} - \eta \nabla_{\theta_{\mathbf{g}}}$
11:     **end for**
12: **end for**

---

Our overall model training algorithm is summarized in Algorithm 1. From Line 2-7, our algorithm is to fix the parameter for $\mathbf{g}$ while updating $\mathbf{f_m}$ and $\mathbf{f_o}$ with a learning rate $\eta$ for $F$ iterations. The gradients w.r.t. $\theta_{\mathbf{f_m}}$ and $\theta_{\mathbf{f_m}}$ are computed as follows:

$$\nabla_{\theta_{\mathbf{f_m}}} = \frac{\partial}{\partial \theta_{\mathbf{f_m}}} \left( \mathcal{L}_{\text{Pred}}(\theta_{\mathbf{f_m}}, \theta_{\mathbf{f_o}}, \theta_{\mathbf{g}}^{\textbf{fixed}}) + \lambda_1 \mathcal{L}_{\text{Param}}(\theta_{\mathbf{f_m}}, \theta_{\mathbf{f_o}}) + \lambda_2 \mathcal{L}_{\text{Feature}}(\theta_{\mathbf{f_m}}, \theta_{\mathbf{f_o}}) \right) \quad (12)$$

$$\nabla_{\theta_{\mathbf{f_o}}} = \frac{\partial}{\partial \theta_{\mathbf{f_o}}} \left( \mathcal{L}_{\text{Pred}}(\theta_{\mathbf{f_m}}, \theta_{\mathbf{f_o}}, \theta_{\mathbf{g}}^{\textbf{fixed}}) + \lambda_1 \mathcal{L}_{\text{Param}}(\theta_{\mathbf{f_m}}, \theta_{\mathbf{f_o}}) + \lambda_2 \mathcal{L}_{\text{Feature}}(\theta_{\mathbf{f_m}}, \theta_{\mathbf{f_o}}) \right) \quad (13)$$

From Line 8-11, our algorithm fixes the parameter for $\mathbf{f_m}$ and $\mathbf{f_o}$ while updating $\mathbf{g}$ with a learning rate $\eta$ for $G$ iterations. The gradients w.r.t. $\theta_{\mathbf{g}}$ are computed as follows:

$$\nabla_{\theta_{\mathbf{g}}} = \frac{\partial}{\partial \theta_{\mathbf{g}}} \left( \mathcal{L}_{\text{Pred}}(\theta_{\mathbf{f_m}}^{\textbf{fixed}}, \theta_{\mathbf{f_o}}^{\textbf{fixed}}, \theta_{\mathbf{g}}) + \lambda_3 \mathcal{L}_{\text{MLP}}(\theta_{\mathbf{g}}) \right) \quad (14)$$

We repeat the overall alternating training process from Line 2-11 stated above for $T$ iterations until the training reaches optimum (e.g., prediction accuracy does not increase on the validation set).

## 4 EXPERIMENTS

**LIDC-IDRI** LIDC-IDRI (Armato III et al., 2011) consists of lung cancer screening thoracic computed tomography (CT) scans with marked-up annotated lesions. We preprocess the 3D nodule images into 2D images by taking the middle slice along the z-axis and keeping the dimension as $224 \times 224$. Annotations are from experienced thoracic radiologists in XML format. After locating the nodules, we further slice surrounding areas as our negative samples. To simulate the incomplete prompt, we randomly add noise by converting pixel values from 0 to 1 outwardly along the annotation boundary. The dataset after preprocessing includes a total of 2625 nodules and 65505 non-nodules images. We use this dataset for the nodule classification task, where the objective is to determine whether an image includes a nodule or not. To better simulate a more practical situation where we only have limited access to the human explanation labels, we randomly sample 100/1200/1200 images for training, validation, and testing and keep the class ratio balanced.

**Pancreas** We obtained the normal pancreas images from Cancer Imaging Archive (Roth et al., 2015)and abnormal images from the Medical Segmentation Decathlon dataset (MSD [1]). The data includes 281 CT scans with tumor v.s. 80 CT scans without tumors. The data preprocessing process is the same as LIDC-IDRI. The MSD dataset includes two types of annotations: tumor lesions and pancreas segmentation. We treat the tumor lesions as our inaccurate explanation labels and pancreas segmentation as our ground truth accurate explanation label. We randomly sample 30 images for training, 30 images for validation from each class, and the rest for testing. The number of samples from each class is balanced in training and testing splits.

---

[1]Available online at: http://medicaldecathlon.com/

Table 1: The prediction evaluation on both datasets on different attention-guided learning methods. The best results for each task are highlighted in boldface font and the second bests are underlined.

| Model | Pancreas | | | | LIDC-IDRI | | | |
|---|---|---|---|---|---|---|---|---|
| | Accuracy ↑ | Precision ↑ | Recall ↑ | F1 ↑ | Accuracy ↑ | Precision ↑ | Recall ↑ | F1 ↑ |
| Baseline | 85.089 | 98.816 | 83.693 | 90.223 | 66.404 | 59.288 | 69.020 | 63.466 |
| GRADIA | 83.132 | 99.042 | 81.12 | 89.103 | 67.435 | 65.645 | 73.184 | 68.991 |
| HAICS | 86.441 | 98.994 | 85.104 | 91.239 | 66.855 | 64.709 | 74.604 | 69.136 |
| RES-G | 89.893 | 98.943 | 89.170 | 93.791 | 68.557 | **67.930** | 71.459 | 69.267 |
| RES-L | 89.786 | 98.065 | 89.876 | 93.784 | 68.353 | 65.966 | **76.275** | 70.551 |
| Proposed | **92.307** | **99.840** | **91.026** | **95.302** | **69.453** | 67.433 | 75.248 | **71.127** |
| Proposed-P | 93.137 | 99.780 | 92.593 | 96.052 | 70.656 | 67.783 | 75.346 | 71.365 |

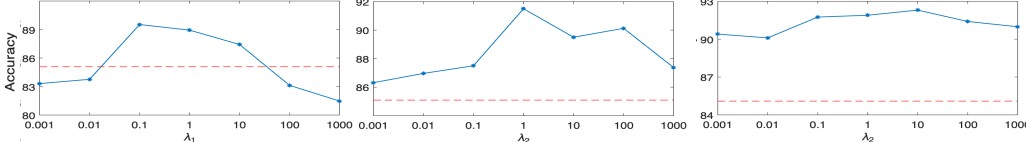

Figure 4: Sensitivity Analysis on the Pancreas dataset.

**Implementation Details** For all methods studied in this work, the CNN model is ResNet18 (He et al., 2016). The batch size, number of perturbed masks $N$, and pixel conversion probability $p$ are set to be 16, 5000, and 0.1, respectively. The models are trained for 30 epochs using the Adam optimizer (Kingma & Ba, 2014) with a learning rate of 0.0001. All experiments are conducted using Amazon EC2 g4dn.xlarge instances, which come equipped with NVIDIA Tesla T4 GPUs.

**Evaluation Metrics** To evaluate the classification performance of proposed models, we use conventional metrics such as accuracy, precision, recall, and F1 score. These metrics provide a comprehensive evaluation of the model's classification capabilities, considering aspects like the balance between precision and recall (F1 score) and overall prediction correctness (accuracy).

**Comparison Methods** In order to assess the efficacy of our proposed framework, we undertake comparative studies with conventional CNN classification models and four prevalent explanation supervision methods, namely, GRAIDA (Gao et al., 2022b), HAICS (Shen et al., 2021), RES-G, and RES-L (Gao et al., 2022c), each distinguished by different implementations of the imputation function. We train these comparison methods following their implementation guidelines. We also compare with a baseline model, which is a ResNet-18 architecture trained exclusively using prediction loss with original images as input. We employ two settings to evaluate our proposed methods: **Propose**, which uses images without prompts for evaluation to align with the implementation settings of comparison methods, and **Propose-P**, which utilizes images with prompts in the testing set for evaluation.

### 4.1 RESULTS EVALUATION

Table 1 shows the quantitative evaluation of two classification tasks. Overall, our method outperforms all other explanation-guided learning methods on both LIDC-IDRI and pancreas datasets. In particular, our model achieves the best accuracy and F1 on the pulmonary nodule classification task and the best accuracy, precision, recall, and F1 on the pancreatic tumor classification task. Our method achieved an accuracy of 92.307% and 69.453% on the two respective datasets, representing an improvement of 8.5% and 4.6% compared to the baseline Resnet-18 approach. In addition, our method attained F-1 scores of 95.302% and 71.127% on the two datasets, marking an enhancement of 5.6% and 12.1% respectively when compared to the baseline Resnet-18 method. Our findings demonstrate that the performance of our prediction methodology is enhanced when incorporating explanations into the prediction phase. While the precision or recall scores reported on the LIDC-IDRI dataset are marginally lower, our approach achieves the highest F1 scores across both datasets. This outcome signifies that our overall performance surpasses that of all other attention-guided learning methods. Also, the strong and consistent performance observed between the proposed and proposed-P demonstrates that our framework can effectively enhance predictions for both images with and without provided attention prompts. As depicted in Figure 5, The attention maps, modified by our proposed attention modification method, are qualitatively shown to be more accurate and fine-grained compared to the initial visual attention prompts with incomplete information. This demonstrates the effectiveness of our attention modification method.

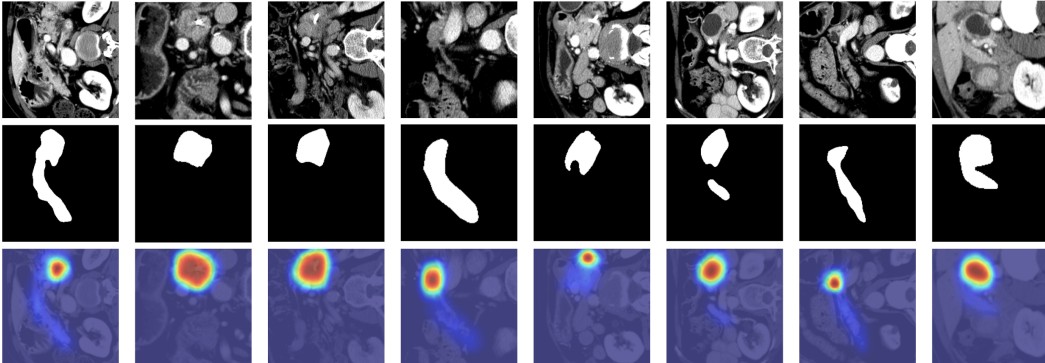

Figure 5: Selected visual prompt modification visualization results for pancreatic cancer classification are presented. Row 1 displays the original images; Row 2 showcases visual attention prompts with incomplete information; Row 3 features the attention map modified using our proposed attention map modification method.

Table 2: Ablation study on LIDC-IDRI (left) and Pancreas dataset (right).

| Model | LIDC-IDRI | | | | Pancreas | | | |
|---|---|---|---|---|---|---|---|---|
| | Accuracy ↑ | Precision ↑ | Recall ↑ | F1 ↑ | Accuracy ↑ | Precision ↑ | Recall ↑ | F1 ↑ |
| Proposed | 69.453 | 67.433 | 75.248 | 71.127 | 92.307 | 99.840 | 91.026 | 95.302 |
| Proposed−**Ours-1** | 67.182 | 64.658 | 75.793 | 69.784 | 89.246 | 98.135 | 85.758 | 91.530 |
| Proposed−**Ours-2** | 66.429 | 63.145 | 72.226 | 67.381 | 88.256 | 96.143 | 81.425 | 88.174 |
| Proposed−**Ours-3** | 68.135 | 64.746 | 73.425 | 68.813 | 90.456 | 98.542 | 89.592 | 93.854 |

We further provide a sensitivity analysis of hyper-parameters $\lambda_1$, $\lambda_2$, and $\lambda_3$, which denotes the weight of parameter sharing regularization loss, feature map sharing regularization loss, and MLP regularization loss, respectively. Figure 4 shows the accuracy of the proposed model for various weights on the pancreatic tumor classification. The red dashed lines represent the baseline model's performance. Overall, the model is sensitive to the variance in weight and we can observe a concave curvature on all three hyper-parameters. The model yields the best overall performance when $\lambda_1$, $\lambda_2$, and $\lambda_3$ are 0.1, 1, and 10, respectively.

## 4.2 ABLATION STUDY

To evaluate the effectiveness of the proposed framework, we conducted an ablation study on four variants of three frameworks: 1) (**Ours−1**) the removal of the visual attention map modification method while directly using attention prompt with incomplete information, 2) (**Ours−2**) the removal of parameter and feature map regularization between the two CNNs during training, and 3) (**Ours−3**) the removal of the proposed weight learning function, using weighted averaging based solely on confidence scores computed by classifiers. The results of the ablation study on both datasets are shown in Table 2. The results of **Ours−3** indicate that the improved predictability of our proposed model is largely due to the regularized two-stream CNN architecture that takes images with and without prompts separately for prediction. From the result of **Ours−1**, we observe that the proposed visual attention map modification method is another key factor contributing to the improved performance.

## 5 CONCLUSION

This research paper introduces a new framework called the Visual Attention-Prompted Prediction and Learning framework. This framework incorporates visual attention prompts into the decision-making process of the model. It effectively addresses the challenges faced in attention-prompted image prediction, such as incomplete information from visual attention prompts and the prediction of samples without prompts. This is achieved by using a regularized two-stream CNN architecture that shares knowledge about prompt information between CNNs and enhances future predictions for images with and without prompts. Additionally, an attention prompt modification method with adaptive mask aggregating is introduced to handle the incompleteness of information for visual attention prompts. The proposed framework is tested on two datasets, and the results confirm its effectiveness in improving the model's prediction for samples with and without attention prompts.

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
