# OpenReview forum: "Visual Attention-Prompted Prediction and Learning"
_ICLR.cc/2024/Conference — ICLR 2024 Conference Withdrawn Submission_

### Official Review · Reviewer_Vm67 · 2023-10-19

**Soundness:** 2 fair
**Presentation:** 3 good
**Contribution:** 2 fair
**Rating:** 3
**Confidence:** 4

**Summary:**

This paper proposes an attention-based distillation method that transfers knowledge between a two-stream convolutional neural network (CNN), where one stream is operating on the original image while the other uses attention-modulated images. The principal idea behind the method is to adaptively integrate a collection of randomly perturbed masks based on their corresponding predictive performance to generate reasonable attention prompts, and encourage alignment between the features and parameters for the two streams. Experimental results on two medical imaging datasets show that the proposed approach is able to outperform baseline and other visual explanation methods.

**Strengths:**

+ It is a promising direction to develop interpretable models with visual attention, especially for applications involving high-stake decisions (e.g., clinical diagnosis).

+ The proposed method does not require the presence of attention prompts during inference, which could potentially accommodate the scarcity of high-quality explanations.

+ The method shows consistent improvements over the baseline on the two selected datasets.

**Weaknesses:**

- The paper ignores a body of research on learning with privileged information (e.g., [ref1, ref2, ref3]), which is closely related to the proposed method in terms of incomplete information during inference. In addition, leveraging a two-stream CNN for distilling privileged knowledge is also not particularly new (e.g., [ref3] also encourages alignment between features from the two streams).

- According to Equation 2, the high-quality attention prompt D is never used to modulate the images in the second stream. Is there a particular reason why always using the perturbed attention prompts?

- Related to the previous comment. I have difficulty understanding the design choice of the perturbed masks. Specifically, Equation 2 discards all important/unimportant pixels (lambda=+1 or -1) and only takes into account regions with unknown importance. What do we want the model to focus on regions with higher uncertainty, instead of those more prominent ones?

- I am a bit concerned about the computational overhead of the model. For each image, it increases the inference by N+1 times (N is the number of perturbed masks). The alternative training also makes the cost complicated. I would suggest the authors provide a quantitative measure of the computational cost.

-  The experiments are inadequate to demonstrate the advantages of the proposed method, mainly because of two reasons: (1) Tasks and scale of the datasets. The paper only evaluates on two small medical imaging datasets for binary classification, and it is unclear if the method can generalize toward broader domains. Moreover, for each dataset, the authors only sample a small subset of data for experiments (e.g., 100 and 30 images for training on the two datasets, respectively). With such a small scale, it is very hard to validate whether or not the method is truly advantageous. (2) Model selection. The paper only tests with ResNet-18, does the method work on other more advanced backbones? I would suggest the authors experiment with state-of-the-art networks designed for medical imaging tasks (or test on ImageNet classification). Alternatively, for a fair comparison with other visual explanation methods, it would also be reasonable to directly test on the same tasks.

- For a more comprehensive evaluation of the explainability of the method, it is reasonable to perform a quantitative evaluation of the model attention to see how they align with expert annotations.

References:

[ref1] Learning Using Privileged Information: Similarity Control and Knowledge Transfer. JMLR, 2015.

[ref2] Unifying Distillation and Privileged Information. ICLR, 2016.

[ref3] Modality Distillation with Multiple Stream Networks for Action Recognition. ECCV, 2018.

**Questions:**

(1) Please elaborate on the key differentiations between the proposed framework and previous works on learning with privileged information.

(2) Please justify the logic behind the design of perturbed masks, especially the questions raised in the weaknesses section.

(3) What is the value of N? What is the computational overhead, e.g., FLOPs of the method when attention prompts are given?

(4) How does the model perform on data of a larger scale? For instance, what if performing the experiments on the full set of LIDC-IDRI? In addition, I would encourage experiments on standard image recognition benchmarks (e.g., ImageNet).

---

### Official Review · Reviewer_HNQQ · 2023-10-26

**Soundness:** 2 fair
**Presentation:** 1 poor
**Contribution:** 2 fair
**Rating:** 3
**Confidence:** 4

**Summary:**

This paper investigates visual attention-guided learning and presents a two-branch framework aimed at improving the model's decision-making process. The proposed framework incorporates various designs, including perturbation-based attention aggregation, attention-prompt guided multi-task learning, and alternating training. The experimental results on LIDC-IDRI and Pancreas datasets support the proposed method.

**Strengths:**

1. The writing is good, and the motivation is sound.
2. The proposed framework demonstrates promising results on the LIDC-IDRI and Pancreas datasets.

**Weaknesses:**

1. The proposed method appears to be overstated. In detail,

- The framework can be seen as a stacked version of two CNN layers, with one leveraging attention annotations while the other does not. In scenarios where attention annotations are not available, it seems that only one branch of the model is necessary, rendering the other redundant.

- The advantages of the proposed attention prompt modification are not clearly conveyed. On one hand, obtaining the modified attention map requires the model to propagate each sample numerous times (equal to the number of augmented images). This can be time-consuming and degrade the efficiency. On the other hand, there have been many methods (e.g., RES) focused on improving noisy attention prompts. Yet, no comparisons are given in this regard.

- The name of “attention-prompt guided multi-task” is confusing. While the authors show that the learning objective includes several components (e.g., classification loss, parameter/feature regularization loss), this should not be treated as “multi-tasks” in my view. Because both network branches still account for the same classification problem.

2. Many design choices are not self-contained. Specifically, why knowledge sharing is necessary for the two CNN networks? Can we just use the prediction from one branch as the final model decision? What are the advantages of this two-branch framework over the one-branch model?

3. The results primarily focus on medical datasets. If this work specifically targets medical analysis, the authors should explicitly state it. Otherwise, it would be crucial to provide additional results on other scenarios (as referenced by RES) to showcase the applicability of the proposed method.

**Questions:**

See Weaknesses.

---

### Official Review · Reviewer_Zfz6 · 2023-11-01

**Soundness:** 3 good
**Presentation:** 3 good
**Contribution:** 2 fair
**Rating:** 6
**Confidence:** 3

**Summary:**

This paper proposes a method that modifies attention maps to enhance a model's predictive power. The model shows strong performance across two well-known datasets.

**Strengths:**

1. The paper is well-written, and the method is clearly explained.

2. The proposed method makes interesting use of a two-stream structure to cope with different kinds of input, i.e. one has image and attention prompt, and the other one only has an image. Although several works have been done in other research fields, this seems novel in this area.

3. The authors also proposed two techniques to further improve the robustness and the accuracy of their models, i.e., the averaged attention map, and the optimized averaging function.

4. The experiments are impressively comprehensive, covering two popular datasets.

5. The authors also consider a number of different ablations to better understand the proposed method.

**Weaknesses:**

1. Though the ablation experiments are extensive, it seems that there may be a very important ablation that was not performed. From my point of view, the $p$ in Eqn (2) is very important for different attention map generation. However, the authors do not take an in-depth study on this hyper-parameter. I think it is important to perform a targeted ablation about that.

2. Why the incomplete input - Proposed is better than Proposed-P in most cases in Tab. 1 ?

3. It seems that three $lambda$ are very sensitive to the final results as shown in Fig. 4. By introducing too many hyper-parameters, this method is not robust.

4. I am also curious to see what the meaning of final learned attention is, is there any change as compared with the ones trained without an attention map?

5. There are no failure cases, which I believe is very important for the reader to understand the proposed technique.

**Questions:**

See Weaknesses

---

### Official Review · Reviewer_5dYi · 2023-11-03

**Soundness:** 2 fair
**Presentation:** 2 fair
**Contribution:** 2 fair
**Rating:** 5
**Confidence:** 3

**Summary:**

The paper addresses the domain of Explanation (attention)-guided learning, focusing on the challenge of enhancing a model’s predictive capabilities by leveraging human-like understanding during training. Traditional approaches in attention-guided learning show promise but are hindered by the need for intensive model retraining, which is both time-consuming and resource-intensive. The paper introduces a framework that significantly reduces the need for model retraining in attention-guided learning. In particular, this work proposes a technique enabling models to utilize attention prompts directly in prediction, bypassing retraining. It also introduces a new framework adept at integrating attention prompts into decision-making and adapting to images with or without such prompts. Regarding methods, this work refine incomplete attention prompts through perturbation-based attention map modification and an innovative optimization-based mask aggregation. The experimental results from two datasets to some extent demonstrate the efficacy of the framework in improving predictions for samples, with or without attention prompts, streamlining the predictive power of attention-guided learning models.

**Strengths:**

This is an interesting and useful work and the merits of the paper are listed below:
1. The problem addressed by the paper is highly relevant. The quest for explainable AI, particularly in the domain of computer vision, is both timely and essential. The work helps bridge the gap between human interpretability and machine learning.
2. The proposed dual-stream CNN architecture is interesting. The proposed perturbation-based method for attention map modification demonstrates a sophisticated approach to dealing with the inherent uncertainty in human-annotated data.
3. The framework’s ability to learn from both prompted and unprompted images without extensive retraining addresses a significant bottleneck in explainable AI.
4. The paper appears to be well-written, easy to follow and understand.

**Weaknesses:**

This work also has the following weaknesses:

1. The method assumes that attention prompts are available or can be generated for all images. However, the process of creating or obtaining these prompts for a vast array of images in real-world datasets could be challenging and is not addressed in detail.
2. The model’s performance seems heavily reliant on the perturbation-based attention prompt map modification. The randomness introduced here and its impact on model stability and consistency over multiple runs is not clearly addressed.
3. There is a potential concern regarding how well the model would generalize to images significantly different from those it was trained on, especially considering that the attention prompts might not be accurate or might introduce bias towards certain features.
4. The model includes several hyperparameters (e.g., the number of perturbed masks $N$, probability $p$, $\lambda_1$, $\lambda_2$, $\lambda_3$). The process of tuning these hyperparameters is not extensively discussed. Despite the experiments shown in Fig. 4, what other insights could there be regarding the hyper-parameter tuning? The performance of the model exhibits sensitivity to the variation in $\lambda_1$, $\lambda_2$, and $\lambda_3$, suggesting that careful tuning is essential for optimal performance.
5. The model's adaptability to other imaging modalities (such as MRI, PET scans, etc.) or different domains is not discussed. This adaptability is vital for the model's utility in a broader context.

**Questions:**

I do not have other questions. The authors may refer to the weaknesses column and see my comments there.